# Oral health outcomes and shift working among male workers: A cross-sectional survey

**Hadi Ghasemi**\*, **Reza Darmohammadi, Mahshid Namdari, Zahra Ghorbani**

Department of Community Oral Health, School of Dentistry, Shahid Beheshti University of Medical Sciences, Tehran, Iran

\* ha.ghasemi@sbmu.ac.ir, hadighasemi558@yahoo.com

## Abstract

Working time outside routine daily working hours is known as shift working. Studies have shown adverse effects of shift working such as stress, obesity, and diabetes on the workers' health. The aim of the present study was to compare aspects of oral health in shift workers and non-shift workers of a cement factory in Shahroud, Iran. Study population comprised of 180 male workers of the factory in the year 2015. Convenience sampling was continued until recruiting 180 subjects of shift- and non-shift workers. Data collection included oral health questionnaire, health and safety executive (HSE) questionnaire, and clinical oral examination. The Chi-square test, Pearson correlation coefficient, and generalized Poisson model were employed for statistical evaluation. Mean age of the workers was 39.19 (±9.48); 53% had educational level of less than diploma. Their mean DMFT was 12.89 (±5.75) which correlated with number of years in shift work schedule (Pearson correlation coefficient: 0.41; $p<0.001$) but not correlated with job stress (Pearson correlation coefficient: -0.11; $p = 0.12$). Mean number of deep periodontal pockets among the workers was 5.03 (±1.84) that showed correlation with number of years in shift work schedule (Pearson correlation coefficient: 0.33; $p<0.001$) but no correlation with job stress (Pearson correlation coefficient: -0.03; $p = 0.68$). Adherence to various oral health behaviors was reported by less than half of the workers. Positive correlation of dental caries and periodontal diseases with shift working partly signals negative impact of working conditions on oral health among this group of workers which calls for modifications in their working environment to facilitate health practices.

## Introduction

Occupation may influence physical and mental health of individuals depending on the type of work and its conditions [1]. Special attention has been paid to the relationship between working hours and the amount of job stress with individuals' health. There is a direct relationship between working hours and damage to health [2]. In addition to the amount, the distribution of working hours over a circadian rhythm also affects health. The effect of shift work schedule on health has been documented [3]. Studies have shown relationship between working in different shifts with several unhealthy conditions such as diabetes [4], cardio-vascular diseases

**Competing interests:** The authors have declared that no competing interests exist.

[5], obesity [6], and metabolic syndrome [7]. Job stress is another factor that affects health both physically and mentally [8].

Occupation can affect oral health through stress or systemic illnesses caused by overwork [9, 10]. In their study on oral health of 3253 office workers, Scalco et al. reported that overwork was associated with poorer oral health status and restricted individuals from adherence to oral health- related behaviors [11]. Youshino et al. found that frequent stomatitis, gingival swelling and recession, slimy feeling in the mouth, bad breath, clicking sound in the jaw, and teeth with attrition were associated with job stress among a group of workers in Japan [12]. In Iran, studies regarding the relationship between shift working and oral health status of workers are, however, limited. In an attempt to examine the relationship between shift working and oral health status of a group of nurses, Tirgar et al. [13] found no statistical significant difference between oral health status of shift working and non-shift working nurses. The aim of this study was to evaluate the impact of working conditions including shift work schedule and job stress on oral health in a group of workers of Shahroud Cement Factory.

## Materials and methods

The target population for this cross-sectional study was 180 male workers of Shahroud Cement Factory in the year 2015. This sample size was estimated based on the findings of a previous study [11] that noticed an approximate 19% difference in the level of stress among two groups of their study population, and considering 80% power and 5% level of significance [14]. They were selected as a convenience sample from total 500 workers of the factory. In a period of twenty consecutive working days, workers were asked to participate and finally 180 workers in different status of shift working accepted to take part in the study. Data collection included questionnaire and clinical oral examination. Questionnaire survey consisted of two sections: 1. Health, oral health, quality of life, and work related questions and 2. Job stress questions.

A group of five professors from the department of community oral health, Shahid Beheshti School of Dentistry checked face and content validity of the questionnaire. Moreover, 15 workers were asked to fill the questionnaire in order to pilot test its validity and to be sure that it is prepared in simple language and in accordance with their level of knowledge. Based on the feedback of the professors and the workers, some minor revisions were applied to the questionnaire. Those 15 workers were also asked to fill the questionnaire again after two weeks in order to determine the reliability of the questionnaire using the test-retest method. The reliability coefficient was 0.7, which is in the acceptable range.

The first section of the questionnaire included questions on the workers' demographic information such as age, years of work experience, working currently in shift work schedule (yes or no), history of shift working (number of years in shift work schedule), having a second job (yes or no), level of education, oral health related behaviors (frequency of toothbrushing, time of last dental visit, frequency of consumption of sugary snacks and drinks), aspects of oral health-related quality of life (frequency of dental pain, restriction of eating and communication with others due to oral problems based on the study of Ghorbani et al. [15]). In order to evaluate the impact of shift work schedule on the workers' oral health, for further analysis, the participants were categorized into three subgroups based on years they have worked under shift work schedule as follows: no history of shift working (0 years), <10 years in shift work schedule, and ≥10 years in shift work schedule.

The second section of the questionnaire included the HSE (Health and Safety Executive) questionnaire [16] which employed to assess job stress. This questionnaire includes 35 questions in seven sub-areas as follows: 1. Demand: issues such as workload, characteristics and work environment, 2. Control: to what extent individuals can be said to be doing their job, 3.

Support of the officials: the amount of support that a person receives from his management and service institution, 4. Peer support: the amount of support that individuals receives from their colleagues, 5. Relationships: "includes promoting positive working to avoid conflict and dealing with unacceptable behavior", 6. Role: "whether people understand their role within the organization and whether the organization ensures that the person does not have conflicting roles", and finally 7. Change: "how organizational change is managed and communicated". The validity and reliability of the Persian version of this questionnaire has been proven in the Iranian population [17]. The answer to each question was based on a 5-point Likert scale from never (score 1) to always (score 5), with higher scores indicating higher job stress. Based on total score (theoretical range: 35–175), the variable job stress was dichotomized into low (<115) and high (≥116) categories.

The Ethics committee of the Shahid Beheshti School of Dentistry approved this study (ethic code: IR.SBMU.RIDS.REC.1394.155). The workers gave their informed consent by accepting to fill the questionnaire. They have given explanation on how to fill the questionnaire, the confidentiality of their answers, and the possibility of withdrawing from the study at each stage. For illiterate workers, the questionnaire, was read and their answers were recorded by one of the authors (RD).

Clinical oral examination included measuring periodontal pocket depth and DMFT index based on the WHO method for oral health survey [18]. One of the authors (RD) who was a senior dental student performed clinical oral examination. He received detailed education about the examination process and reached to an acceptable level of agreement with an experienced clinical professor. Dental caries was recorded at the level of cavity and periodontal pocket with >4 mm depth (deep periodontal pocket) was considered as a sign of periodontal disease.

The SPSS software version 22 was employed for the statistical evaluation which included: the Chi- square test (for assessing the difference between subgroups of the workers based on their shift- work schedule and background, job related, and oral health related variables), Pearson correlation coefficient (for assessing bivariate correlation between dental caries and periodontal disease with shift-work schedule and job stress), and generalized Poisson model (for the multivariate analysis of factors contributing to the level of DMFT and periodontal pockets).

## Results

The workers' mean age was 39.19 (±9.48) with average 13.91 (±7.24) years of work experience and 35% reported ≥10 years in shift work schedule. As it appears in Table 1, majority of the workers reported that working in the cement factory is their only job and for more than half of the workers, education level was below diploma and perceived job stress was high.

Table 2 shows distribution of the workers' oral health-related behaviors and aspects of oral health–related quality of life according to their shift work pattern. The respondents' adherence to the oral health-related behaviors was as follows: at least twice per day toothbrushing by 9% (at least once per day by 18%), using dental floss at least once per day by 1%, eating or drinking sugary foods at least once per day by 42%, and having a dental visit in the past year by 33%. More than three forth of the respondents reported that they had rarely or never pain, difficulty in communication, and difficulty in eating due to dental problems in the past three months.

As it appears in Table 2, among the present workers, oral health-related behaviors and quality of life showed no statistical significant correlation with their history of shift working.

Mean of DMFT among the workers was 12.89 (±5.75), that showed a correlation with number of years in shift work schedule (Pearson correlation coefficient: 0.41; p<0.001) but no

**Table 1. Distribution (%) of the workers (n = 180), based on their background and working factors.**

| | | All (%) | Years in shift work schedule | | |
| --- | --- | --- | --- | --- | --- |
| | | | < 10 (%) | ≥ 10 (%) | P-value* |
| Age (years) | ≤ 35 | 62 (34) | 54(47) | 8 (13) | <0.001 |
| | 36–50 | 97 (54) | 56 (48) | 41 (64) | |
| | ≥ 51 | 21 (12) | 6 (5) | 15 (23) | |
| Working experience (years) | < 15 | 93 (52) | 74 (64) | 19 (30) | <0.001 |
| | ≥ 15 | 87 (48) | 42 (36) | 45 (70) | |
| Having second job | No | 155 (86) | 102 (88) | 53 (83) | 0.37 |
| | Yes | 25 (14) | 14 (12) | 11 (17) | |
| Education level | < Diploma | 96 (53) | 59 (51) | 37 (58) | 0.43 |
| | ≥ Diploma | 84 (47) | 57 (49) | 27 (42) | |
| Job stress level | Low | 86 (48) | 54 (47) | 32 (50) | 0.75 |
| | High | 94 (52) | 62 (53) | 32 (50) | |

* Statistical evaluation by the Chi-square test.

correlation with job stress (Pearson correlation coefficient: -0.11; p = 0.12). Mean number of periodontal pockets with >4mm depth among the participants of this study was 5.03 (±1.84). This variable also showed a correlation with number of years in shift work schedule (Pearson correlation coefficient: 0.33; p<0.001) but no correlation with job stress (Pearson correlation coefficient: -0.03; p = 0.68).

Table 3 presents factors which were interested to evaluate their effect on the expected count of DMFT and the number of sites with periodontal disease among the workers based on two similar models. In these models, shift work schedule and job stress seem to have no impact on DMFT and periodontal pocket.

**Table 2. Distribution (%) of the workers' (n = 180) self-reported oral health-related behaviors and quality of life based on their history of shift working.**

| | | All (%) | Years in shift work schedule | | | |
| --- | --- | --- | --- | --- | --- | --- |
| | | | 0 (%) | < 10 (%) | ≥ 10 (%) | P-value[1] |
| Toothbrushing | ≥2/day | 16 (9) | 3 (9) | 8 (9) | 5 (8) | 0.93 |
| | <2/day | 164 (91) | 29 (91) | 76 (90) | 52 (92) | |
| Using dental floss | ≥1/day | 2 (1) | 0 (0) | 1 (1) | 1 (2) | 0.78 |
| | <1/day | 178 (99) | 32 (100) | 83 (99) | 63 (98) | |
| Dental visit in the past 12 months | Yes | 60 (33) | 11 (34) | 28 (33) | 21 (33) | 0.98 |
| | No | 120 (67) | 21 (66) | 56 (67) | 43 (67) | |
| Sugary snack or drink between main meals | ≤1 | 75 (42) | 13 (41) | 37 (44) | 25 (39) | 0.82 |
| | >1 | 105 (58) | 19 (59) | 47 (56) | 39 (61) | |
| Dental pain in the past 3 months | Seldom[2] | 141 (78) | 23 (72) | 69 (82) | 49 (77) | 0.44 |
| | Regularly | 39 (22) | 9 (28) | 15 (18) | 15 (23) | |
| Difficulty in communication due to dental problems in the past 3 months | Seldom[2] | 164 (91) | 31 (97) | 76 (90) | 57 (89) | 0.43 |
| | Regularly | 16 (9) | 1 (3) | 8 (9) | 7 (11) | |
| Difficulty in eating due to dental problems in the past 3 months | Seldom[2] | 145 (81) | 26 (81) | 70 (83) | 49 (77) | 0.58 |
| | Regularly | 35 (19) | 6 (19) | 14 (17) | 15 (23) | |

1. Statistical evaluation by the Chi-square test.

2. Seldom: rarely or never; regularly: sometimes, mostly or always.

**Table 3. Determinants of dental caries and periodontal disease among the workers (n = 180) as assessed by two similar generalized Poisson models.**

| | IRR | SE | *p*-value | 95% CI |
|---|---|---|---|---|
| **DMFT** | | | | |
| Shift work schedule (0 = no, 1 = yes) | 1.003 | 0.003 | 0.399 | 0.99–1.01 |
| Age (years) | 1.027 | 0.002 | <0.001 | 1.02–1.03 |
| Education level (0 = <Diploma, 1 = ≥ Diploma) | 1.037 | 0.047 | 0.428 | 0.94–1.13 |
| Job stress score | 0.999 | 0.002 | 0.805 | 0.99–1.00 |
| Have a second job (0 = no, 1 = yes) | 1.066 | 0.069 | 0.327 | 0.93–1.21 |
| Frequency of dental visit in the past year | 0.972 | 0.022 | 0.218 | 0.93–1.01 |
| Frequency of tooth brushing | 0.940 | 0.014 | **<0.001** | **0.91–0.96** |
| Frequency of using dental floss | 1.118 | 0.070 | 0.074 | 0.98–1.26 |
| Constant | 5.432 | 1.743 | | |
| **Number of sites with periodontal pocket >4mm** | | | | |
| Shift work schedule (0 = no, 1 = yes) | 1.005 | 0.003 | 0.141 | 0.99–1.01 |
| Age (years) | 1.011 | 0.002 | <0.001 | 1.00–1.01 |
| Education level (0 = <Diploma, 1 = ≥ Diploma) | 1.029 | 0.045 | 0.516 | 0.94–1.12 |
| Job stress score | 1.000 | 0.002 | 0.958 | 0.99–1.00 |
| Have a second job (0 = no, 1 = yes) | 1.051 | 0.065 | 0.421 | 0.93–1.18 |
| Frequency of dental visit in the past year | 0.999 | 0.022 | 0.978 | 0.95–1.04 |
| Frequency of toothbrushing | 0.968 | 0.013 | **0.018** | **0.94–0.99** |
| Frequency of using dental floss | 1.123 | 0.059 | **0.028** | **1.01–1.24** |
| Constant | 2.867 | 0.893 | | |

IRR: incidence rate ratio, SE: standard error.

## Discussion

Findings of the present study failed to show a relationship between job stresses and oral health outcomes among the workers. A weak correlation, however, was found between shift work schedule with higher scores of DMFT and higher number of deep periodontal pockets in the bivariate analysis. Favorable level of dental floss and toothbrushing was rare among the workers in spite of high emphasis on these basic self-care behaviors for maintaining satisfactory level of oral health [19, 20].

Psychological stress in result of working condition has been suggested as a risk factor for dental caries and periodontal diseases. One of the probable mechanisms in this regard is the effect of psychological stress on the individual's oral health behaviors [21]. Among the workers in the present study, work-related stress showed relationship neither with measures of dental caries and periodontal disease, nor with oral health related behaviors and oral health related quality of life. Studies on the relationship between job stress and measures of oral health, however, show some sort of controversy. While some studies found that higher level of work stress is associated with poor oral health related quality of life [12, 22], and less favorable level of oral health related behaviors [23], a systematic review found no evidence on the association of work stress with dental caries and tooth loss but some potential association between periodontal disease and work stress [24].

Being in the shift schedule for a long time correlated with less favorable oral health status among the workers in the present study in a way that those workers with more years in shift schedule experienced higher level of dental caries and periodontal diseases. When adjusting for the other covariates affecting dental caries and periodontal disease in multivariate analysis

by means of generalized Poisson models, the history of working in shift work schedule showed no impact on dental caries and periodontal disease in the workers. This may be due to the effect of age variable since history of shift work for more than 10 years might simply mean that the individual is more aged and therefore reporting more age related dental problems. A positive association between shift-working and higher prevalence of dental caries and periodontal disease, has, also, been evidenced in two studies on workers in Turkey [25] and Japan [26]. This finding is in line with studies showing the association of non-standard work schedules with less favorable health outcomes [27–29]. Non-standard working schedule is proposed as a trigger for physiological changes resulting to disturbances in sleep pattern and circadian rhythms and ultimately reduce the workers' adaptive capabilities [30]. The relationship between various working time characteristics with employees' health status has been studied largely. For example, young adult American workers without a regular day shift pattern of working showed higher risk for a wide range of health-related outcomes comparing to workers with standard regular daytime schedule [28]. Irregular work schedule found to have negative impact on quality of sleep and sense of coherence among a group of Hungarian nurses [29]. Due to the importance of oral health on the quality of life and productivity of the workers, incorporating the oral health indicators into the comprehensive workers' health surveys is called for.

Adherence to healthy behaviors is an essential prerequisite for good oral health status. Very few workers in this study have, however, reported complying with toothbrushing at least twice per day and using dental floss. Regular dental visit and restriction of sugary food were also a rare finding since about two third of the workers did not report a dental visit and close to 60% reported more than once per day eating sugary foods. This may be a reflection of either under-estimation of the importance of best oral hygiene practices by those workers or their difficult working condition that hinder them from performing recommended level of oral health behaviors. Data on the oral health of the workers in Iran is rare. However, findings of a recent study on a group of employees in Tehran [31], shows higher prevalence for toothbrushing (28%), using dental floss (48%), and dental visit (77%) compared to the present study. Another study from Mashhad, Iran [32], found that at least 34% of the employees brush their teeth twice per day and 35% use dental floss in a daily basis. Different figures of the prevalence of oral health behaviors reflect underestimation of the importance of recommended criteria for favorable level of oral health and also different level of socio-economic status and cultural diversity of the participants in the previous studies. Reaching to higher rates of favorable level of oral health behaviors among working age population is not a dream since findings of some studies from foreign countries show promising results in this regards. For example, tooth-brushing on at least 2/day basis has been reported by more than 90% of a group of Korean workers [33] and around 80% of a representative sample of adult population from eight European countries [34].

Workers' health is exposed to the risks resulting from different working conditions, considering that they generally spend a relatively long time in the workplace. Therefore, it is the strength of this study that investigated about the impact of occupational conditions on the oral health of this target group. Findings of this study should, however, be interpreted with caution due to some limitations. First is the inherent limitation of cross-sectional study which cannot make a causal inference between variables. Second is the convenient sampling which makes the generalization of the findings to the whole population uncertain. Third is the limitation of questionnaire surveys which is prone to socially acceptable answers and recall bias. Conducting longitudinal studies with control group on a representative sample of workers whose health status and working conditions are registered routinely would, therefore, provide better evidence.

## Conclusions

Positive correlation between years in shift working with dental caries and periodontal diseases in this study speaks for a potential risk of non-standard working time for oral health of the workers. Provision of rather stable working time would help workers for better adherence to optimal oral health behaviors which in turn enhance their oral health and overall health status and ultimately their productivity in their work environment.

## Supporting information

**S1 File.**
(SAV)

## Acknowledgments

We thank all the participating workers for their cooperation.

## Author Contributions

**Conceptualization:** Hadi Ghasemi, Zahra Ghorbani.

**Data curation:** Reza Darmohammadi, Mahshid Namdari, Zahra Ghorbani.

**Formal analysis:** Hadi Ghasemi, Mahshid Namdari, Zahra Ghorbani.

**Investigation:** Reza Darmohammadi.

**Methodology:** Hadi Ghasemi, Mahshid Namdari, Zahra Ghorbani.

**Project administration:** Hadi Ghasemi, Reza Darmohammadi.

**Supervision:** Hadi Ghasemi.

**Validation:** Hadi Ghasemi.

**Writing – original draft:** Hadi Ghasemi, Reza Darmohammadi.

**Writing – review & editing:** Hadi Ghasemi, Mahshid Namdari, Zahra Ghorbani.

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
