## [Decision Letter · Decision Letter 0]

23 Jun 2022

PONE-D-22-05469The relationship between oral health and occupational condition in workers of a cement factoryPLOS ONE

Dear Dr. Hadi Ghasemi,

Thank you for submitting your manuscript to PLOS ONE. After careful consideration, we feel that it has merit but does not fully meet PLOS ONE’s publication criteria as it currently stands. Therefore, we invite you to submit a revised version of the manuscript that addresses the points raised during the review process.

ACADEMIC EDITOR:Dear Dr Hadi Ghasemi, it is indeed a interesting manuscript which you have submitted. However, the reviewers have certain queries regarding your manuscript. I request you to answer those queries.==============================

We look forward to receiving your revised manuscript.

Kind regards,

Tanay Chaubal

Academic Editor

PLOS ONE

Journal Requirements:

Reviewers' comments:

Reviewer's Responses to Questions

**Comments to the Author**

1. Is the manuscript technically sound, and do the data support the conclusions?

Reviewer #1: Yes

Reviewer #2: No

2. Has the statistical analysis been performed appropriately and rigorously? 

Reviewer #1: No

Reviewer #2: Yes

3. Have the authors made all data underlying the findings in their manuscript fully available?

Reviewer #1: No

Reviewer #2: No

4. Is the manuscript presented in an intelligible fashion and written in standard English?

Reviewer #1: Yes

Reviewer #2: No

5. Review Comments to the Author

Reviewer #1: This is a good paper on an interesting topic. However, there are certain concerns that I would like to have the authors address,

Methods:

Comment 1: In what language was the questionnaire administered? If it was translated from English, was a linguistic expert used for the translation and back translation of the questionnaire?

Comment 2: Was a standard questionnaire used for assessing OHR-QoL? If not, why was a standard instrument not used?

Comment 3: How was the sample size determined for convenience sampling?

Comment 4: Please describe the complete statistical analysis in detail along with the software used for the analysis and purpose of each specific test in the last section of the methods.

Results

Comment 4: The study's objective is to compare shift with non shift workers, why then is years of shift work schedule compared in Table 1 and Table 2? It would be more appropriate to compare shift with non-shift workers.

Comment 5: Shift work for more than 10 years might simply mean that the individual is more aged and therefore reporting more age related dental problems, unless the age factor is adjusted for this association is questionable.

Reviewer #2: English revision required.Authors have gone through the surface only, kindly go deep in to the subject and modify the article. Some findings are missing, some are contradictory to your own conclusions. Kindly do the modifications

kindly see the PDF for all comments

6. PLOS authors have the option to publish the peer review history of their article (what does this mean?). If published, this will include your full peer review and any attached files.

Reviewer #1: No

Reviewer #2: **Yes: **AMITHA BASHEER N

---

## [Author Response · Author response to Decision Letter 0]

23 Aug 2022

Response to Reviewers

PONE-D-22-05469

Reviewer #1: This is a good paper on an interesting topic. However, there are certain concerns that I would like to have the authors address,

Methods:

Comment 1: In what language was the questionnaire administered? If it was translated from English, was a linguistic expert used for the translation and back translation of the questionnaire?

Answer: the questionnaire was administered in Persian. The process of translation and back translation by a linguistic expert has been conducted in a previous study [Azad ME, Gholami FM. Reliability and validity assessment for the HSE job stress questionnaire. QUARTERLY INTERNATIONAL JOURNAL OF BEHAVIORAL SCIENCES: WINTER 2011, Volume 4, Number 4 (14)] as It has been noted in the last sentence of the 4th paragraph of the Method section that “The validity and reliability of the Persian version of this questionnaire has been proven in the Iranian population (13).”. 

Comment 2: Was a standard questionnaire used for assessing OHR-QoL? If not, why was a standard instrument not used?

Answer: Questions regarding OHR-Qol were based on the following article: “Ghorbani Z, Ahmady AE, Ghasemi E, Zwi A. Socioeconomic inequalities in oral health among adults in Tehran, Iran. Community Dent Health. 2015 Mar 1;32(1):26-31.” Which has now been added to the end of 3rd paragraph of Method section.

Comment 3: How was the sample size determined for convenience sampling?

Answer: In this regard, an explanation has now been added to the first paragraph of the method section as follows:” Based on the findings of a previous study (Scalco GP, Abegg C, Celeste RK, Hökerberg YH, Faerstein E. Occupational stress and self-perceived oral health in Brazilian adults: a Pro-Saude study. Ciência & Saúde Coletiva. 2013;18:2069-74.), that noticed an approximate 19% difference in the level of stress among two groups of their study population, and considering 80% power and 5% level of significance (Dhand, N. K., & Khatkar, M. S. (2014). Statulator: An online statistical calculator. Sample Size Calculator for Comparing Two Independent Proportions. Accessed 22 July 2022 at http://statulator.com/SampleSize/ss2P.html), we estimated 90 workers for each group of the current shift- and non-shift workers.”

Comment 4: Please describe the complete statistical analysis in detail along with the software used for the analysis and purpose of each specific test in the last section of the methods.

Answer: the last paragraph of the method has, now, been changed as follows: “The SPSS software version 22 was employed for the statistical evaluation which included: the Chi-square test (for assessing the difference between subgroups of the workers based on their shift-work schedule and background, job related, and oral health related variables), Pearson correlation coefficient (for assessing bivariate correlation between dental caries and periodontal disease with shift-work schedule and job related stress), and generalized Poisson model (for the multivariate analysis of factors contributing to the level of DMFT and periodontal pockets).”

Results

Comment 4: The study's objective is to compare shift with non shift workers, why then is years of shift work schedule compared in Table 1 and Table 2? It would be more appropriate to compare shift with non-shift workers.

Answer: For more clarification, some revisions made of the introduction (last sentence: The aim of this study was to evaluate the impact of working conditions including shift work schedule and job stress on oral health in a group of workers of Shahroud Cement Factory ), Method (4th paragraph: “…working currently in shift work schedule (yes or no), history of shift working (number of years in shift work schedule),…” and “In order to evaluate the impact of shift work schedule on the workers’ oral health, Ffor further analysis, the participants were categorized into three subgroups based on years they have worked under shift work schedule as follows: no history of shift working (0 years), <10 years in shift work schedule, and ≥10 years in shift work schedule.”), and Tables 1, 2 (one column representing no history of shift working was added).

Comment 5: Shift work for more than 10 years might simply mean that the individual is more aged and therefore reporting more age related dental problems, unless the age factor is adjusted for this association is questionable.

Answer: the following sentence has, now, been added to the discussion (3rd paragraph) which further explain this issue: “When adjusting for the other covariates affecting dental caries and periodontal disease in multivariate analysis by means of generalized Poisson models, the history of working in shift work schedule showed no impact on dental caries and periodontal disease in the workers. This may be due to the effect of age variable since history of shift work for more than 10 years might simply mean that the individual is more aged and therefore reporting more age related dental problems.”.

Reviewer #2: 

English revision required. 

Answer: a comprehensive English revision has, now, been performed throughout the text.

Authors have gone through the surface only, kindly go deep in to the subject and modify the article. 

Answer: principal changes have, now, been took place on different parts of the manuscript to meet this comment.

Some findings are missing, some are contradictory to your own conclusions. Kindly do the modifications. 

Answer: Contradictory findings with conclusions have, now, been modified in first and 3rd paragraph of the discussion. Findings with higher priority were decided to be presented.

kindly see the PDF for all comments:

1. Title can be more specific and clear. Mention about study design/study setting/or variables to be measured in the title.

Title has now been changed to “Oral health outcomes and shift working among male workers: A cross-sectional survey”.

2. Kindly be specific about which occupational condition

In the running title, “Occupational condition” has now been substituted with “shift working”.

3. Kindly follow the journal guidelines to for headings

Now, headings have been changed according to journal guidelines.

4. Where it is used? In the body of the article, there is no mention regarding it.

“WHO standard” has now been excluded from the sentence “WHO standard oral health questionnaire….”in the abstract. 

5. Try to change these keywords with more relevant ones.

They have now been changed as follows: Shift work schedule, oral health, occupational stress, Dental caries, Periodontal disease

6. Please follow journal guidelines for citation. Use square parentheses

Now, square parentheses have been used. 

7. Repetition of previous line. Modify the sentence

The sentence has now been modified as follows: “In addition to the amount, the distribution of working hours over a circadian rhythm also affects health.”

8. Why 9.5? Please justify

It has now been corrected.

9. Assessed

The sentence has now been changed to: “In their study on oral health of 3253 office workers, Scalco et al. reported that…..”.

10. What is this unusual feeling? Please make it clear. Modify and rewrite the sentence

“unusual feeling” has been changed to “slimy feeling” in the sentence.

11. Use standardized terms " attrition"

“…and worn down teeth were associated…” has now been changed to “…and teeth with attrition were associated…”.

12. Limited or no studies are there? If yes, kindly quote that study

One study has now been quoted as follows: “In an attempt to examine the relationship between shift working and oral health status of a group of nurses, Tirgar et al. (2016) (Tirgar Aram, Mohebbi Simin Zahra, Shaneie Fereshteh, Nikpour Maryam, Parhiz Alireza. The Relationship of Shift Work And Oral Health In Nurses. JOURNAL OF ERGONOMICS FALL 2016 , Volume 4 , Number 3) found no statistical significant difference between oral health status of shift working and non-shift working nurses”.

13. Modify the sentence and rewrite it

The sentence “Data gathering continued until reaching 90 workers from each of the shift work schedule or not shift work schedule (stable shift) in a period of twenty consecutive working days.” has now been changed to “In a period of twenty consecutive working days, workers were asked to participate and finally 180 workers in different status of shift working accepted to take part in the study.”.

14. How did the authors checked content validity of the questionnaire. please mention about it .

The 2nd paragraph of the Method section has now been modified as follows: “A group of five professors from the department of community oral health, Shahid Beheshti School of dentistry checked face and content validity of the questionnaire. Moreover, 15 workers were asked to fill the questionnaire in order to pilot test its validity and to be sure that it is prepared in simple language and in accordance with their level of knowledge. Based on the feedback of the professors and workers, some minor revisions were applied to the questionnaire. Those 15 workers were also asked to fill the questionnaire again after two weeks in order to determine the reliability of the questionnaire using, the test-retest method. The reliability coefficient obtained at this stage was 0.7, which is in the acceptable range.”

15. participants were divide in to two groups at first. Then when did they divided into these groups. This part doesn't have clarity. please mention clearly.

Answer: the sentence has now been changed as follows: “Based on total score (theoretical range: 35-175), the variable job stress was dichotomized into low (<115) and high (≥116) categories.”

16. What about informed consent? Is oral consent is enough in this age group?

Answer: the sentence has now been changed as follows: “The workers gave their informed consent by accepting to fill the questionnaire”.

17. Who did the clinical examination?any training for the examiner was given? Mention Cronbach's alpha

Answer: The sentence has now been changed as follows: “One of the authors (RD) who was a senior dental student performed clinical oral examination. He received detailed education about the examination process and reached to an acceptable level of agreement with an experienced clinical professor.”.

Since the process of calibration was a part of routine education of the whole dental students (including RD), the Cronbach’s alpha did not registered.

18. What about the illiterate people?

Answer: “Education level” has now been categorized in three groups to include “illiterate” as well in the Table 1.

19. Please modify table 2

Answer: it has now been modified completely.

20. Make the title more specific and simple

Answer: title of table 2 has now been modified as follows: “Distribution (%) of the workers’ (n=180) self-reported oral health-related behaviors and quality of life based on their history of shift working.”.

21. Then what about 90 non shift workers? Table title shows n=180

Answer: the total 180 workers has now been categorized into three groups based on their reported years in shift working to cover non-shift workers as well.

22. total number of periodontal pocket? or number of participants ?

Answer: “Mean number of periodontal pockets with >4mm depth among the participants of this study was 5.03 (±1.84).” means that: on average, each participant had 5.03 periodontal pockets with >4mm depth.

23. Modify table 3, add columns or rows and make it more attractive and legible

Answer: it has now been modified. Both columns and rows have been added.

24. Please modify and rewrite discussion. needs English revision Most of the findings of the study weren't discussed. Give proper justification for the findings

Answer: the discussion part has now been modified thoroughly with comprehensive English revision and more justification for the findings.

25. First and second sentences are contradictory

Answer: it has now been revised. (Also see below)

26. What authors want to convey through this sentence?

Answer: the whole paragraph has now been changed as follows: “Findings of the present study failed to show a relationship between job stresses with oral health outcomes among the workers. A weak correlation, however, was found between shift work schedule and higher scores of DMFT and higher number of deep periodontal pockets in the bivariate analysis. favorable level of dental floss and toothbrushing was rare among the workers in spite of high emphasis on these basic self-care behaviors for maintaining good satisfactory level of oral health (Benzian H, Williams D. The challenge of oral disease: a call for global action. The oral health atlas. 2nd ed. Geneva: FDI World Dental Federation 2015. https:// www. fdiworlddental.org/sites/default/files/media/documents/complete_oh_atlas.pdf

Davies RM, Davies GM, Ellwood RP, Kay EJ. Prevention. Part 4: Toothbrushing: What advice should be given to patients? Br Dent J 2003Aug; 195:135-41.).”.

27. Please explain

Answer: the explanation has come immediately in the next sentence as follows: “While some studies found that higher level of work stress is associated with poor oral health related quality of life (16) (11)., and less favorable level of oral health related behaviors(17), a systematic review found no evidence on the association of work stress with dental caries and tooth loss but some potential association between periodontal disease and work stress (18).”.

28. Please check journal guidelines and write the citations

Answer: the citations have now been changed based on the journal guideline.

29. Strengths of the study? 

Answer: the following sentence has now been added to the beginning of the last paragraph of the discussion part: “Workers’ health is exposed to the risks resulting from different working conditions, considering that they generally spend a relatively long time in the workplace. Therefore, it is the strength of this study that investigated about the impact of occupational conditions on the oral health of this target group.”.

30. Recommendations?

Answer: the following sentence has now been added to the end of the last paragraph of the discussion part as follows: “Conducting longitudinal studies with control group on a representative sample of workers whose health status and working conditions are registered routinely would, therefore, provide better evidence.”.

31. Doesn't match with study findings

Answer: the conclusion has now been revised as follows: “Positive correlation between years in shift working with dental caries and periodontal diseases in this study speaks for a potential risk of non-standard working time for oral health of the workers. Provision of rather stable working time would help workers for better adherence to optimal oral health behaviors which in turn enhance their oral health and overall health status and ultimately their productivity in their work environment.”.

---

## [Decision Letter · Decision Letter 1]

26 Sep 2022

The relationship between oral health and occupational condition in workers of a cement factory

PONE-D-22-05469R1

Dear Dr. Hadi Ghasemi,

We’re pleased to inform you that your manuscript has been judged scientifically suitable for publication and will be formally accepted for publication once it meets all outstanding technical requirements.

Kind regards,

Tanay Chaubal

Academic Editor

PLOS ONE

Additional Editor Comments (optional):

Reviewers' comments:

Reviewer's Responses to Questions

**Comments to the Author**

1. If the authors have adequately addressed your comments raised in a previous round of review and you feel that this manuscript is now acceptable for publication, you may indicate that here to bypass the “Comments to the Author” section, enter your conflict of interest statement in the “Confidential to Editor” section, and submit your "Accept" recommendation.

Reviewer #1: All comments have been addressed

Reviewer #2: All comments have been addressed

2. Is the manuscript technically sound, and do the data support the conclusions?

Reviewer #1: Yes

Reviewer #2: Yes

3. Has the statistical analysis been performed appropriately and rigorously? 

Reviewer #1: Yes

Reviewer #2: Yes

4. Have the authors made all data underlying the findings in their manuscript fully available?

Reviewer #1: Yes

Reviewer #2: (No Response)

5. Is the manuscript presented in an intelligible fashion and written in standard English?

Reviewer #1: Yes

Reviewer #2: Yes

6. Review Comments to the Author

Reviewer #1: (No Response)

Reviewer #2: Article has been modified nicely and clearly. All comments have been dressed.

All the very best. Thank you.

7. PLOS authors have the option to publish the peer review history of their article (what does this mean?). If published, this will include your full peer review and any attached files.

Reviewer #1: No

Reviewer #2: **Yes: **AMITHA BASHEER N

---

## [Editor Report · Acceptance letter]

7 Oct 2022

PONE-D-22-05469R1 

Oral health outcomes and shift working among male workers: A cross-sectional survey 

Dear Dr. Ghasemi:

I'm pleased to inform you that your manuscript has been deemed suitable for publication in PLOS ONE. Congratulations! Your manuscript is now with our production department. 

Kind regards, 

on behalf of

Dr. Tanay Chaubal 

Academic Editor

PLOS ONE